# Can Extended Chemotherapy Improve Glioblastoma Outcomes? A Retrospective Analysis of Survival in Real-World Patients

**DOI:** 10.3390/jpm12101670

**Published:** 2022-10-08

**Authors:** Natalia Gherasim-Morogai, Vlad-Adrian Afrasanie, Bogdan Gafton, Mihai Vasile Marinca, Teodora Alexa-Stratulat

**Affiliations:** 1Medical Oncology Department, Regional Institute of Oncology, 700483 Iasi, Romania; 2Oncology Department, Faculty of Medicine, Grigore T. Popa University of Medicine and Pharmacy, 700115 Iasi, Romania

**Keywords:** glioblastoma, prognosis, chemotherapy, adjuvant temozolomide, number of cycles

## Abstract

Standard treatment for glioblastoma multiforme (GBM) is surgery followed by radiotherapy plus concurrent chemotherapy with daily temozolomide (TMZ), and six subsequent TMZ 5/28-day cycles. Research has focused on identifying more effective alternatives to the current protocol, including extension of the number of adjuvant TMZ cycles. We performed a retrospective analysis of all GBM patients treated in our hospital (160 patients, 2011–2020). Median follow-up was 16.0 months. Analysis of prognostic factors was performed with a particular focus on the benefit of extending TMZ chemotherapy. Improved survival correlated with younger age, female gender, good performance status, absence of cognitive dysfunctions, no steroid use, and total tumor resection. Median progression-free survival (PFS) was 12 months and median overall survival (OS) was 20.0 months for the entire cohort. Median OS by adjuvant TMZ was 10.0 months if no adjuvant chemotherapy given (group 0), 15.0 months for patients that did not complete six TMZ cycles (group A), 24.0 months for those that did (group B), and 29.0 months for patients having received more than six cycles (group C) (*p* < 0.0001). At the three-year mark, 15.9% patients were alive in group A, 24.4% in group B and 38.1% in group C. Carefully selected GBM patients may derive benefit from extending the standard adjuvant chemotherapy beyond six TMZ cycles, but more data is required.

## 1. Introduction

Glioblastoma multiforme (GBM) is the most frequent primary brain tumor, with an incidence rate of 0.6–3.7 per 100,000 persons depending on geographical area [1,2]. Current trends indicate a steady increase in the number of new cases over the last 30 years [3], while the slight male predominance [4] and the 64-year median age at diagnosis remain constant [2]. Standard treatment for operable GBM is maximal surgical resection followed by concurrent external beam radiation therapy (EBRT, given as a total dose (TD) of 60 Gy in 2-Gy daily fractions, 5 days a week) and continuous chemotherapy with temozolomide (TMZ) for 42 days. TMZ is an oral alkylating prodrug, whose metabolite (MTIC) has the ability to introduce methyl groups into DNA, thus inhibiting its replication. Due to its small size (194 Da) and lipophilic nature, TMZ can penetrate the blood–brain barrier and is FDA-approved for the treatment of aggressive central nervous system cancers such as anaplastic astrocytoma and GBM [5]. After the completion of EBRT, patients receive an additional six adjuvant TMZ 5/28-day cycles This treatment protocol has been the cornerstone of GBM management ever since the results of the randomized Phase III trial by Stupp et al. were published in 2005 and showed a 2.5-month survival benefit compared to prior standard treatment [6]. However, GBM remains a cancer with a high fatality-to-case ratio, with a median overall survival (mOS) of 14–17 months [7] and a five-year survival rate of approximately 10% in clinical trials and 3–4% in the real-world setting [8].

Several patient and tumor characteristics (younger age at diagnosis, small tumor size, gross tumor resection, a good performance status, and female gender) have been correlated with increased survival in various studies. Seeking to create a unitary prognosis tool, Curran et al. developed the recursive partition analysis (RPA) classification through statistical analysis of a large glioma database. In its first version, the RPA classification included all primary brain tumors and grouped patients according to age, histology, performance status, mental status, and type of surgery into one of six prognostic classes. Survival rates progressively decreased across RPA classes, with class I RPA having the best (mOS 58.6 months), and class VI RPA the worst prognosis (mOS 4.6 months) [9]. This tool has been since improved and refined to include GBM-only patients and has been validated across several GBM patient populations [10,11]. More recently, two additional molecular prognosis factors have been identified—the IDH1/2 mutation and the methylguanine-DNA methyltransferase (MGMT) promoter methylation [12]. However, despite improvements in molecular biology, prognostic assessment, and understanding of pathophysiology, no significant novel treatments have been approved for first-line GBM since 2005. As such, several attempts have been made to increase survival by modifying the Stupp protocol, e.g., administering higher TMZ doses [13] or extending the duration of adjuvant TMZ beyond six cycles [14]. The reasoning behind this approach is that TMZ-induced DNA alkylation most often leads to cell apoptosis, senescence, or autophagy, especially in the setting of low MGMT levels [5]. Other theoretical arguments supporting a higher number of TMZ adjuvant cycles include good tolerance of the drug, with few serious grade adverse events reported [15] and the lack of effective second line treatments [16].

Published data on the effectiveness of this approach yield controversial results. Due to the low frequency of GBM, its high fatality rate and the relative lack of interest for an “old” drug, real-world data and retrospective analyses remain key factors for improving GBM patient management and for validating prognostic and predictive factors that can help with clinical decision-making.

The aim of our study is to determine overall survival (OS) and progression-free survival (PFS) of a large cohort of consecutive GBM patients treated by means of surgery, radiation therapy and TMZ chemotherapy in a tertiary oncology center. Furthermore, we retrospectively analyzed the impact of prognostic factors on survival, with a particular focus on the potential benefits of extending adjuvant TMZ treatment length beyond the standard six cycles.

## 2. Materials and Methods

We performed a retrospective analysis of all GBM patients treated in the Regional Institute of Oncology Iasi—a 330-bed reference center for cancer patients in North-East Romania (roughly 5 million inhabitants)—over a period of ten years (1 January 2011 to 31 December 2020). The study was approved by the local Ethics Committee; due to its retrospective nature, rigorous anonymization and reporting of aggregate data, no specific patient consent was required.

All patients were adults (age over 18) with a confirmed diagnosis of GBM, treated by surgery (total or subtotal resection) and adjuvant EBRT (with or without concurrent chemotherapy) within 90 days after the initial diagnosis. We excluded patients with low-grade gliomas, those that did not receive both systemic and local treatment or have received postoperative treatment in other centers, and patients enrolled in clinical trials. After analyzing all GMB cases, we found 160 cases that respected the above-mentioned criteria.

Clinical, pathology, and treatment-related information was extracted from patient charts. We collected data on gender, age, residence, ECOG performance status (PS), Mini-Mental State Examination (MMSE) score, neurologic symptoms, corticosteroid use, tumor location and size, IDH1 mutation and MGMT promoter methylation status (where known), imaging method used for EBRT planning, extent of resection, local and systemic treatment details, tumor response, and toxicity—until patient death or date of last follow-up. Each patient included in the analysis was assigned to a RPA class. Gross total resection was defined as the complete removal of tumor(s), as gauged by preoperative and post-operative MRI or CT; all other types of resections were considered subtotal. Information regarding patient survival was obtained from the National Population Registry. Treatment complications and adverse events were graded and assessed by the attending physician using the Common Terminology Criteria for Adverse Events (CTCAE v5.0) and recorded in each patient’s file as per institutional protocol.

Treatment response was assessed by imaging performed at approximately five weeks after EBRT. Results were classified as complete response (CR), partial response (PR), stable disease (SD), and progressive disease (PD) according to the Macdonald criteria, that also take into account some potential confounders such as corticosteroid use [17]. Increased contrast enhancement on MRI scans or cerebral edema that occurs during or after treatment can mimic early tumor progression; when identified, this finding was termed pseudoprogression (PPG) and had to be confirmed as either SD or PD at the next disease assessment (8 weeks later) as per study protocol.

Progression-free survival (PFS) was defined as the time from diagnosis (date of biopsy/resection) to tumor progression or death, as reported by the treating oncologist. Overall survival (OS) was calculated from time of diagnosis to death of any cause, and censored at the pre-specified database lock date.

Descriptive statistics, demographic, and clinical variables were analyzed using the Microsoft Office Excel 2019 for Windows software (Redmond, WA: Microsoft Corp.). All other statistical analyses were performed in SPSS Statistics for Windows, version 25.0 (Armonk, NY: IBM Corp.). Categorical variables were analyzed by means of the Pearson χ2 test or Fisher’s exact test as appropriate. Continuous variables were assessed with the aid of the Student t-test or the Mann-Whitney U test. Estimations of PFS and OS in the overall population and in different subgroups were obtained through the Kaplan–Meier method. P-values comparing the resulting curves were calculated with log-rank tests and were considered statistically significant if less than 0.05. Parameters found to be statistically relevant to PFS and/or OS in univariate analysis were then included in a Cox multinomial regression model, to assess their impact on adjuvant chemotherapy prolongation as an independent prognostic factor for survival. The cut-off date for this analysis was December 2021.

## 3. Results

Database search identified 160 patients that met the pre-specified inclusion criteria and had started treatment in the Regional Institute of Oncology between 1 January 2011, and 31 December 2020.

### 3.1. Patient and Tumor Characteristics

The median age at diagnosis was 56.0 years (range 20–80 years). There was a slight predominance in males (54%, 86/160), with a male to female ratio of 1.16:1. Most patients (55.6%, 89/160) maintained a good performance status (ECOG PS 1) after surgery/biopsy. However, 18 patients (11.2%) had an ECOG of 3 or 4 before the start of radiation therapy. We also identified 21 patients (13.1%) with cognitive dysfunction (MMSE < 27) prior to the start of anti-cancer treatment. Neurologic symptoms were present in 79 patients (49.4%) prior to the start of anti-cancer treatment.

Mean tumor size as assessed before surgery was 48.96 mm (range 18–96 mm). The most frequent tumor localizations were in the frontal (18.8%) and parieto-occipital (15.6%) lobes. Nine patients (5.6%) were diagnosed with multifocal GBM. Only 59 patients (36.9%) underwent complete macroscopic tumor resection. IDH1 mutation status was available for only 24 of the 160 cases (15.0%). MGMT promoter methylation status was not available for any of the patients.

All cases were assigned to RPA III, IV, or V prognostic groups, with most patients (113/160, 70.6%) in the RPA IV class. Additional patient and tumor characteristics can be found in Table 1.

### 3.2. Post-Surgical Treatment

Most of the patients (78.7%) received the standard full-course EBRT protocol for GBM (total dose (TD) = 60 Gy) and were able to complete radiation therapy as prescribed (94.4%) (Table 2). Magnetic resonance imaging (MRI) was used in 99 of the 160 patients (61%) for developing the treatment plan. Over 85% of the patients underwent concurrent chemotherapy with temozolomide (TMZ) and most of them (71.3%) were able to finish the prescribed 42-day course. Adjuvant TMZ was recommended for 127 patients at the end of EBRT. Approximately two-thirds (n = 83) of these received the entire standard course (6 cycles) from which a further 42 patients went on to receive a prolonged chemotherapy course (median number of additional TMZ cycles was 6, range 1–7).

Treatment was discontinued during radiation therapy or during adjuvant chemotherapy in 86 patients (53.7%), most often due to disease progression or toxicity. Seven patients died while receiving radiation therapy and thus were not available for post-EBRT assessments. Generally, both the ECOG scores and cognitive status were maintained, with no statistical differences before vs. after anti-cancer treatment for most patients, while some (11.2%) reported an improvement of their neurological symptoms after EBRT (Table 2). An increase in the number of patients requiring corticosteroids was noted at the end of EBRT (45 patients compared to 19 patients that required corticosteroids before EBRT).

Treatment-related toxicities were recorded in 119/160 patients during EBRT ± concurrent TMZ and in 86/127 patients during adjuvant TMZ. The most frequent side effects of any grade reported during the concurrent part of the treatment were lymphopenia (55.6%), nausea (22.5%), and serum gamma-glutamyl transferase (GGT) increase (21.8%). Grade 3–4 toxicities were recorded in 22 of these patients and mostly consisted in lymphopenia (13.1%) and neutropenia (3.7%). Of note, there was one treatment-related death, as assessed by the treating physician—the patient developed a fatal pulmonary thromboembolism while receiving EBRT (Table 3). During adjuvant TMZ treatment, the most frequent toxicities were lymphopenia (63.7%), serum GGT increase (25.2%), and anemia (22.8%). Grade 3–4 adverse events mostly consisted in lymphopenia (10.2%) and GGT increase (3.1%) and occurred in 14 patients. No grade 5 toxicities were recorded during adjuvant chemotherapy (Table 3).

Imaging assessment (performed five weeks after completion of EBRT) showed that most patients (92 patients, 57.5%) achieved stable disease, with 8 (5.0%) complete responders and 14 (8.7%) partial responders (Table 2). Second-line treatment options for patients with progressive disease were chemotherapy (TMZ, carboplatin or nitrosourea), a second surgery or tumor re-irradiation.

### 3.3. Survival Outcomes

At the time of data cut-off, 20 patients (12.5%) were alive, and 6 (3.7%) were lost to follow-up. Median PFS of the cohort was 12.0 months (95% CI, 9.7–14.2), and median OS was 20.0 months (95% CI, 17.2–22.7); 118 patients (73.8%) reached the 1-year OS landmark, with 65 (40.6%) and 35 patients (21.8%) still alive at 2 and 3 years, respectively.

In univariate analysis, age at diagnosis was inversely correlated with the probability of survival. Patients under the age of 65 had a median PFS of 13.0 months (95% CI, 10.5–15.5) and a median OS of 22.0 months (95% CI, 19.6–22.4), as opposed to 9.0 months (95% CI, 7.1–10.9, *p* = 0.02) and 14.0 months (95% CI, 11.3–16.7, *p* = 0.01), respectively, for patients older than 65 years. Female patients had statistically significant better outcomes, with a median PFS of 15.0 months (95% CI, 11.7–18.3) versus 10.0 months (95% CI, 8.1–11.9) in males (*p* = 0.02), whereas median OS was 24.0 months (95% CI, 20.3–27.7) versus 16.0 months (95% CI, 12.2–19.8) in males (*p* = 0.002).

There was a significant correlation between survival and the pre-treatment ECOG performance status (*p* < 0.0001). Patients with ECOG PS 0 had the longest median PFS (29.0 months; 95% CI, 12.7–45.3) and OS (35.0 months; 95% CI, 21.8–48.2), whereas patients with ECOG PS 1 had a median PFS of 13.0 months (95% CI, 9.5–16.5) and a median OS of 23.0 months (95% CI, 21.1–24.9); lowest OS was recorded for ECOG PS 3 patients (10.0 months; 95% CI, 6.3–13.6). There were no significant differences between pre- and post-treatment median values of ECOG PS score.

The presence of cognitive dysfunction had a negative impact on both PFS and OS. Median PFS was 13.0 months (95% CI, 10.9–15.1) and median OS was 22.0 months (95% CI, 19.2–24.8) in patients with MMSE scores ≥ 27, versus 7.0 (95% CI, 4.3–9.7) and 11.0 months (95% CI, 6.5–15.5), respectively, in patients who scored less than 27 points on the MMSE test (*p* < 0.0001 for both PFS and OS).

Most patients did not receive long-term corticosteroids after surgery or during adjuvant treatment. However, those that did require symptom control after finishing chemo-radiotherapy had a shorter PFS (8.0 months vs. 13.0 months, *p* = 0.003) and a statistically significant shorter median OS (23.0 months vs. 11.0 months, *p* < 0.001) compared to those not on chronic steroid treatment.

A significantly statistical difference in PFS was observed between RPA groups, with a median of 21.0 months (95% CI, 11.6–30.4) for RPA III patients, 12.0 months (95% CI, 9.5–14.5) for RPA IV and 6.0 months (95% CI, 3.6–8.4) for RPA V patients (*p* < 000.1) (Figure 1). This also translated to median OS reaching 35.0 months (95% CI, 28.8–41.2), 20.0 months (95% CI, 16.6–23.4), and 9.0 months (95% CI, 4.2–13.8) for each respective class (*p* < 0.0001) (Figure 2).

Predictably, median PFS was longer in cases where surgery resulted in total resection (13.0 months; 95% CI, 10.5–15.4) as compared to subtotal resection (10.0 months; 95% CI, 7.9–12.0), although this difference was not statistically significant (*p* = 0.19). However, the OS benefit for total resection was clearer (23.0 months; 95% CI, 20.5–25.4, vs. 17.0 months; 95% CI, 13.9–20.0; p = 0.03).

Patients receiving the full radiation dose (60 Gy) and the concurrent TMZ treatment without dose reductions or interruptions had an improved survival irrespective of age group.

### 3.4. Standard Versus Extended Adjuvant TMZ

We divided patients according to the number of adjuvant TMZ cycles received as follows: group A—received less than six cycles (treatment discontinued due to toxicity, progressive disease, or patient preference), group B—received exactly six cycles of adjuvant TMZ (standard treatment), and group C—received more than six cycles of chemotherapy (extended treatment). The decision to extend adjuvant TMZ was made upon consultation with the patient, on a case-by-case basis. General criteria used were: good performance status (ECOG PS 0–1), no new neurological symptoms, no need for corticosteroids, good treatment tolerance (hematologic, gastro-intestinal), no clinically significant disease progression. While this was a retrospective study, the remarkably well-balanced patient disposition between these three groups should be noted (Table 4).

Median PFS was 6 months (95% CI, 3.55–8.44) for patients not receiving chemotherapy, 8.0 months (95% CI, 6.4–9.6) for patients that did not finish the standard adjuvant chemotherapy, 14.0 months (95% CI, 11.0–17.0) for patients completing all the 6 cycles, and 20.0 months (95% CI, 16.4–23.6) for patients who went on to receive more than six cycles. While unwarranted due to the study design, we did look for meaningful differences between the groups receiving standard adjuvant chemotherapy, whether they completed it or not (1–6 cycles) and those who went on to receive more than six cycles (7–12 cycles), and found a significant difference of median PFS of 10 months (10.0 months; 95% CI, 7.94–12.05 vs. 20.0 months; 95% CI, 16.4–23.6, respectively; *p* = 0.006) (Figure 3).

Median OS was 10.0 months (95% CI, 7.6–12.4) if no adjuvant treatment was given, 15.0 months (95% CI, 10.7–19.3) for patients that did not finish six TMZ adjuvant cycles, 24.0 months (95% CI, 20.0–26.0) for patients that finished 6 cycles and 29.0 months (95% CI, 23.1–34.9) for patients that received more than six cycles.

The comparison of the “standard” group (1–6 cycles) with the “extended” group (7–12 cycles) showed a 9-month difference in mOS (20.0 months; 95% CI, 15.6–24.4 vs. 29.0 months, 95% CI 23.1–34.9); this difference was statistically significant, with a P-value of 0.022 (Figure 4). At the one-year mark, 61.4% patients in group A, 82.9% patients in group B and 100% patients in group C were alive. The survival advantage of patients in group C was maintained at two-year and three-year assessment as well—57.1% and 38.1% patients were alive in group C versus 43.9% and 24.4% in group B and 29.5% and 15.9% in group A.

To assess the impact of several confounders identified in univariate analysis on the possible effect of protracted adjuvant chemotherapy on the outcome of GBM, we performed a multivariate regression analysis. The variables included in the model besides the number of adjuvant chemotherapy cycles (≤6 vs. >6 cycles) were gender, age group (<65 years vs. ≥65 years), RPA class, resection (total vs. partial) tumor size (<30 mm vs. ≥30 mm), initial ECOG PS, initial MMSE score (<27 vs. ≥27), use of corticosteroids before treatment (yes vs. no), completing EBRT and concurrent TMZ as prescribed (yes vs. no). The model was robust for both PFS (overall Chi-square value 29.492, *p* = 0.002) and OS (overall Chi-square value 48.103, *p* < 0.0001), and showed that patients who received more than the standard six adjuvant chemotherapy cycles had a 2.3-fold (*p* = 0.008) and 2.6-fold (*p* = 0.006) lower probability of progression and death, respectively (Figure 5).

## 4. Discussions and Conclusions

The present database analysis identified 160 glioblastoma patients that received radiation therapy and chemotherapy in our department. Patient characteristics were similar to those reported in the Stupp landmark study [6]. Of note, there were more patients with cognitive dysfunction at baseline in the Stupp study (29.1%) than in our analysis (13.1%), possibly due to the retrospective nature of the research. Moreover, in the Stupp trial [6], less than 40% of patients (36.5%, 105/287) were able to finish the full adjuvant TMZ course, whereas in the current analysis the rate was over 50% (51.8%, 83/160), possibly due to advances in radiotherapy and supportive care in the past 15 years.

In our study, median PFS was 12 months and median OS was 20 months for the entire cohort. In the Stupp trial, median PFS was 6.9 months and median OS was 14.6 months [6]. Data from other retrospective series and clinical trials report significant variations in OS and PFS of GBM patients, ranging from 4 to 20 months for PFS and 12 to 28 months for OS [3,13,14,15,18,19]. This high variability of retrospective data can be explained by geographical and regional differences in terms of GBM management and the inherent patient selection bias; also, more recent database analyses usually report better survival compared to older reports. Variability in clinical trials can be explained by different inclusion criteria and the small number of patients usually enrolled in GBM trials due to the rarity and aggressiveness of the tumor.

Younger age, female sex, a good performance status, initial lack of cognitive deficit (normal MMSE), total resection, and ability to receive the standard Stupp protocol correlated with improved PFS and OS in our study (univariate analysis), which is why they were included in the multivariate analysis. However, the literature data regarding the impact of these prognostic factors is still controversial. Even though most studies agree on the importance of total resection in GBM management [20,21], there have been some retrospective large database analysis that did not reach the same conclusion [22]. Similarly, performance status has been identified as an independent prognostic factor for survival in most studies (including our analysis), but not all data support these findings [23]. Similar controversies are noted regarding the importance of age or gender in terms of GBM survival—some authors did not find any correlation between gender and survival [3,20] or age and survival [24], whereas others [4,19] have underlined their importance. This can be partly explained by differences in creating age categories—some studies use the 50-year cut-off, while others use the 60-year [3] or the 70-year cut-off [24].

The RPA groups integrate several of these prognostic factors and thus offer a clearer distinction between good, intermediate, and poor-prognosis patients. As such, almost all studies and retrospective analysis that used RPA grouping have found significant survival differences [3,13,25]. In our study, most patients were considered to be a RPA class IV, which means age < 50 and KPS < 90 OR age ≥ 50 and partial or total resection with no worse than minor neurofunction impairment. Our findings are consistent with data from other studies where most GBM patients are RPA class IV as well, albeit in some studies the percentages are somewhat lower—60.7% RPA IV in the RTOG 0525 study [25] and 46.3% in the GBM retrospective analysis by Ben Nasr et al. [3]

Only 15% of our patients had a known IDH1 mutation status, and no statistical analysis could be performed to determine the impact on survival. MGMT promoter methylation status is not currently part of standard assessments in Romania due to reimbursement issues. Current literature data indicate that both IDH1 and MGMT methylation are independent prognostic factors in glioblastoma. IDH1 testing is performed by immunohistochemistry and mutations are more commonly found in secondary glioblastomas, where they are associated with increased patient survival [26]. MGMT promoter methylation leads to epigenetic silencing of the MGMT gene and is assessed by means of specific sequencing techniques such as methylation-specific polymerase chain reaction (MSP) or pyrosequencing [27]. The impact of MGMT promoter status on survival is still controversial, but available data indicate that patients with methylated MGMT are more likely to respond to chemotherapy and, therefore, have increased survival [28]. However, IDH1 and MGMT status assessment has entered clinical practice in the past ten years and due to the additional costs, these tests are still not performed for each patient. Additionally, a significant number of cases in the current analysis were diagnosed before 2017, when IDH1 status was not assessed in Romania. However, because IDH1 and MGMT status was not tested in most patients, it is very likely that decisions pertaining to disease management (standard versus extended TMZ) were not influenced by these factors.

There has been a constant debate about the best way to improve the Stupp protocol for GBM patients. The first approach considered was increasing TMZ dose. The RTOG 0525 study was a large phase III trial that aimed to determine if a dose-dense (DD) schedule of TMZ can improve outcome in GBM patients [13]. 833 patients were randomized to receive at the end of concurrent chemo-radiotherapy either standard TMZ chemotherapy (Stupp protocol) or the DD protocol that consisted of 75 mg/m2 TMZ for 21 consecutive days of a 28-day cycle, which was increased for subsequent cycles to 100 mg/m2 if no treatment-related adverse events greater than grade 2 were noted. The authors found no difference between arms in terms of PFS or OS, with a median PFS of 5.5 months in the standard arm and 6.7 months in the DD TMZ arm and a median OS of 16.6 months in the standard arm and 14.9 months in the DD TMZ arm. The academic community accepted the results of the RTOG 0525 promptly, and the strategy of using a dose-dense TMZ regimen was abandoned. The second approach tested was extending adjuvant TMZ administration beyond the six cycles of the Stupp protocol. There are some case series that reported over 7 years of continuous TMZ treatment in GBM patients with good tolerance, few side effects, and a satisfactory quality of life [29]. Over time, several other institutions have reported their experience with prolonging TMZ administration.

A retrospective analysis (2004–2014) performed on adult GBM patients treated with radiation therapy and TMZ had similar findings to the present analysis. The authors identified 213 GBM patients and reported a median OS of 15 months and a median PFS of 8 months, with longer OS for patients with a good performance status (ECOG = 0) that underwent gross tumor resection and received adjuvant TMZ beyond six cycles. The authors found that the maximum benefit on OS was noted after more than twelve cycles of TMZ [18]. Another retrospective analysis published by Huang et al. [20] compared patients that received six cycles of adjuvant TMZ (group S) to those that received extended TMZ (group E) and found a median PFS of 15 months in group S and 20.1 months in group E, results which are similar to ours (14 months in the six cycle TMZ group and 20 months in the extended TMZ group). However, contrary to our findings, the authors did not find a statistically significant benefit on OS, even though there was a numerical superiority (median OS was 19.4 months in group S and 25.6 months in group E), most likely due to the small number of patients included in the analysis (approximately 25 patients/group). A slightly larger retrospective study (75 patients) compared standard versus extended TMZ administration and found that OS was almost double (20.6 months versus 47 months) in the extended TMZ group [15]. Of note, the median number of cycles received in the extended TMZ group was 12. An interesting retrospective analysis performed by Seiz and colab. [30] reviewed all GBM cases in their institution, but, contrary to most studies, all patients received TMZ adjuvant therapy until disease progression or unacceptable toxicity, i.e., there was no prespecified number of cycles. The authors concluded that both time to progression (TTP) and OS are directly correlated to the number of cycles administered. It is important to note that per institutional protocol there were GBM patients treated with as many as 57 cycles of adjuvant TMZ.

The GEINO 14-01 randomized multicenter phase II trial was specifically designed to identify potential benefits of extending TMZ adjuvant treatment beyond six cycles [14]. A total of 159 patients with no progressive disease at the end of the concurrent radio-chemotherapy phase of treatment were randomized to receive either six or twelve TMZ adjuvant cycles. The authors found no statistically significant difference in 6-months PFS between groups and no OS benefit for extended TMZ administration. However, the overall sample size was small [31] and almost 40% of patients in the experimental arm did not receive all six additional adjuvant TMZ cycles, as opposed to most retrospective data reviewed above and the data from our study where the median number of cycles was 12. Additionally, authors from the GEINO 14-01 study reported a statistically significant difference between subsequent treatment options—more patients in the experimental arm received no active anti-cancer treatment at progression compared to patients in the standard arm [14]. Nonetheless, this study is very important for GBM research since it is among the only ones that recruited patients in a prospective manner. However, it is important to note that most studies that found extended TMZ treatment benefit administered a median of six or more additional cycles. Barbagallo et al. reported their real-world clinical experience with GBM patients and identified a significant PFS and OS benefit for patients receiving extended TMZ (28 vs. 8 months OS in extended vs. standard treatment patients), but the mean number of cycles administered in the extended TMZ group was 27 [19].

However, when analyzing retrospective data, the choice of standard versus extended TMZ is not random. In such cases, the decision to administer more than six TMZ cycles is usually reserved for selected patients that have responded well to treatment, with little to no side effects, have maintained clinical benefit, and a good performance status, and are willing to continue chemotherapy despite potential risks. Personalizing treatment in GBM patients is very important, especially since there are no new drugs available for first-line treatment, clinical trials are scarce, and survival remains low. Physicians must take into account several factors when recommending a certain course of treatment and the current analysis suggests that in selected cases extending TMZ beyond six cycles can still be a reasonable option.

## Figures and Tables

**Figure 1 jpm-12-01670-f001:**
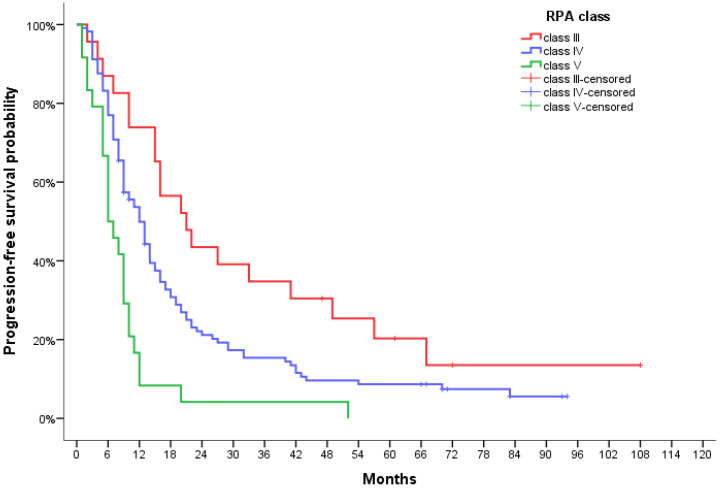
Progression-free survival curve for GBM patients according to RPA class.

**Figure 2 jpm-12-01670-f002:**
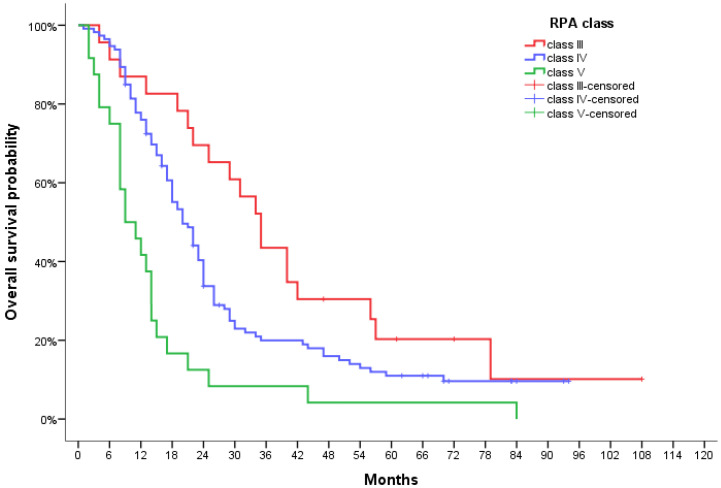
Overall survival curve for GBM patients according to RPA class.

**Figure 3 jpm-12-01670-f003:**
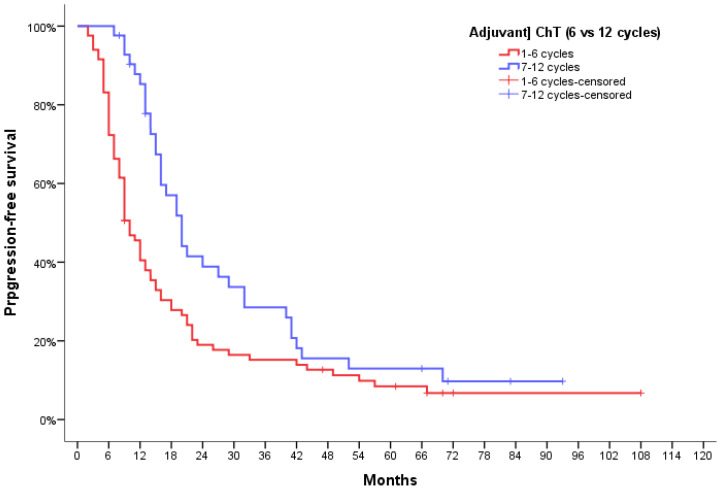
Progression-free survival curves for GBM patients according to the number of adjuvant TMZ cycles received.

**Figure 4 jpm-12-01670-f004:**
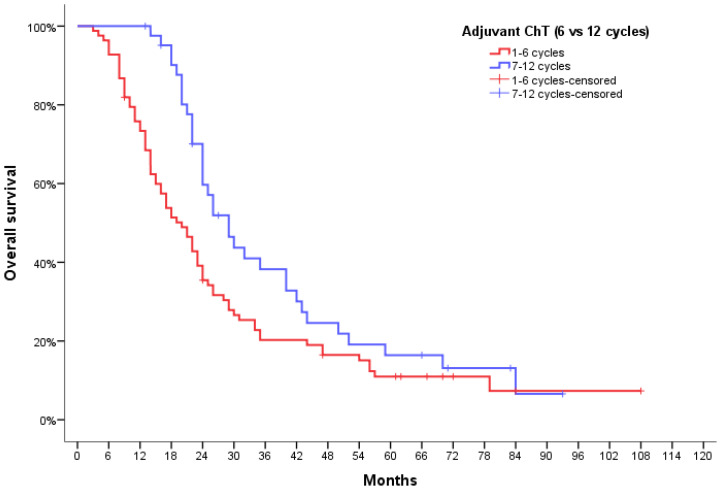
Overall survival curves for GBM patients according to the number of adjuvant TMZ cycles received.

**Figure 5 jpm-12-01670-f005:**
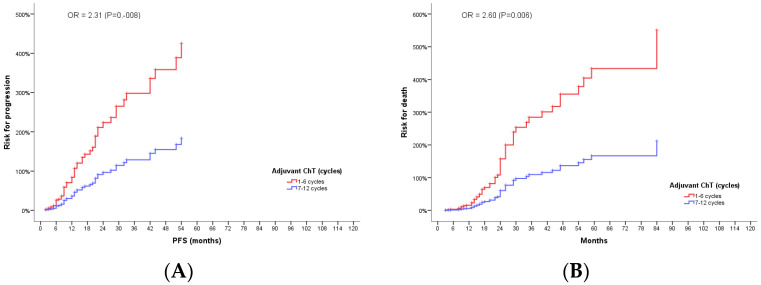
Cox regression model: odds radios for PFS (**A**) and OS (**B**) curves for GBM patients according to the number of adjuvant TMZ cycles received.

**Table 1 jpm-12-01670-t001:** Patient and tumor characteristics.

Parameter	Variable	Number of Patients (N) = 160, n (%N)
Age group	Median [range], years 18–50 51–60 61–70 ≥70	56.0 (20–80) 44 (27.5%) 51 (31.8%) 46 (28.7%) 19 (11.8%)
Gender	Male Female	86 (53.7%) 74 (47.3%)
Residence	Urban Rural	81 (50.6%) 79 (49.4%)
ECOG PS at baseline	0 1 2 3 4	22 (13.8%) 89 (55.6%) 31 (19.4%) 14 (8.8%) 4 (2.5%)
MMSE at baseline	≥27 <27	139 (86.9%) 21 (13.1%)
Neurologic symptoms at baseline	Yes No	79 (49.4%) 81 (50.6%)
Corticosteroid use	Yes No	19 (12%) 141 (88%)
Tumor size	Mean [range], mm	48.96 (18–96)
Tumor localization	Frontal Fronto-temporal Temporal Temporo-parietal Parieto-occipital Other Missing data	30 (18.8%) 16 (10%) 23 (14.4%) 21 (13.1%) 25 (15.6%) 26 (16.2%) 19 (11.9%)
Tumor sites	Unifocal Multifocal	151 (94.4%) 9 (5.6%)
Type of resection	Total Subtotal	59 (36.9%) 101 (63.1%)
IDH1 mutation status	Mutant Wild-type Unknown	9 (5.62%) 15 (9.38%) 136 (85%)
RPA class	III IV V	23 (14.4%) 113 (70.6%) 24 (15%)

List of abbreviations: ECOG: Eastern Cooperative Oncology Group, MMSE: Mini–Mental State Examination, RPA: recursive partitioning analysis.

**Table 2 jpm-12-01670-t002:** Treatment characteristics.

Parameter	Variable	Number of Patients (N) = 160, n (%N)
EBRT TD	60 Gy 40 Gy 25 Gy or less	126 (78.7%) 21 (13.2%) 13 (8.1%)
Imaging method used for planning	MRI CT	99 (61%) 61 (39%)
EBRT finished as prescribed	Yes No	151 (94.4%) 9 (5.6%)
Concurrent TMZ	Yes No	138 (86.3%) 22 (13.7%)
Concurrent TMZ finished as prescribed	Yes No No concurrent TMZ	114 (71.3%) 24 (15%) 22 (13.2%)
Adjuvant TMZ	Yes No	127 (79.4%) 33 (20.6%)
Number of adjuvant TMZ cycles	>6 6 <6 No adjuvant TMZ	42 (26.3%) 41 (25.6%) 44 (27.5%) 33 (20.6%)
Tumor response	CR PR SD PD PPG NA *	8 (5%) 14 (8.7%) 92 (57.5%) 19 (11.9%) 8 (5%) 19 (11.9%)
ECOG PS (post-EBRT)	0 1 2 3 4 NA	24 (15%) 89 (55.6%) 25 (15.6%) 12 (7.5%) 3 (1.9%) 7 (4.4%)
MMSE score (post-EBRT)	≥27 <27 NA	130 (81.2%) 23 (14.4%) 7 (4.4%)
Evolution of neurological symptoms (post-EBRT)	No symptoms Improved Stationary Worsened NA *	81 (50.6%) 18 (11.2%) 44 (27.5%) 10 (6.3%) 7 (4.4%)

*List of abbreviations*: EBRT: external beam radiation. therapy, TD, total dose, MRI: magnetic resonance imaging, CT: computerized tomography, TMZ: temozolomide, CR: complete response, PR: partial response, SD: stable disease, PD: progressive disease, PPG: pseudoprogression, NA: * not available for assessment, ECOG: Eastern Cooperative Oncology Group, MMSE: Mini–Mental State Examination.

**Table 3 jpm-12-01670-t003:** Most frequent treatment-related toxicities.

Adverse Event, N = 160 Patients	EBRT ± Concurrent TMZ (n = 160)		Adjuvant TMZ (n = 127)			
			≤6 Cycles (n = 83)		>6 Cycles (n = 44)	
	*Any,* n (%N)	*Grade 3–4,* n (%N)	*Any,* n (%N)	*Grade 3–4,* n (%N)	*Any,* n (%N)	*Grade 3–4,* n (%N)
Lymphopenia	89 (55.6%)	21 (13.1%)	51 (40.1%)	13 (10.2%)	20 (15.7%)	0 (0%)
Neutropenia	28 (17.5%)	6 (3.7%)	11 (8.6%)	2 (1.6%)	2 (1.6%)	0 (0%)
Febrile neutropenia	6 (3.7%)	0 (0%)	0 (0%)	0 (0%)	0 (0%)	0 (0%)
Anemia	18 (11.2%)	1 (0.6%)	16 (12.5%)	0 (0%)	5 (3.9%)	0 (0%)
Thrombocytopenia	32 (20%)	3 (1.8%)	16 (12.5%)	1 (0.7%)	4 (3.1%)	0 (0%)
Infection	17 (10.6%)	0 (0%)	0 (0%)	0 (0%)	0 (0%)	0 (0%)
Rash	6 (3.7%)	0 (0%)	0 (0%)	0 (0%)	0 (0%)	0 (0%)
ALAT/ASAT increase	15 (9.3%)	3 (1.8%)	9 (7.0%)	2 (1.6%)	4 (3.1%)	2 (1.6%)
GGT increase	35 (21.8%)	4 (2.5%)	15 (11.8%)	2 (1.6%)	7 (5.5%)	1 (0.7%)
Hyperglycemia	7 (4.3%)	0 (0%)	0 (0%)	0 (0%)	0 (0%)	0 (0%)
Nausea	36 (22.5%)	1 (0.6%)	18 (14.1%)	0 (0%)	6 (4.7%)	0 (0%)
Vomiting	4 (2.5%)	1 (0.6%)	3 (2.3%)	0 (0%)	4 (3.1%)	0 (0%)
Constipation	3 (1.8%)	1 (0.6%)	0 (0%)	0 (0%)	0 (0%)	0 (0%)
Thromboembolism	3 (1.8%)	2 * (1.2%)	0 (0%)	0 (0%)	0 (0%)	0 (0%)
Confusion	8 (5%)	1 (0.6%)	0 (0%)	0 (0%)	0 (0%)	0 (0%)

*List of abbreviations*: N: total number of patients in the study; n: number of patients experiencing side effects; GGT: gamma-glutamyl transferase; ALAT: alanine aminotransferase; ASAT: aspartate aminotransferase; TMZ: temozolomide. * one of the two cases was a grade 5 thromboembolism.

**Table 4 jpm-12-01670-t004:** Distribution of study population according to adjuvant treatment group.

Adjuvant Chemotherapy (TMZ) N = 127	Standard Adjuvant TMZ Not Finished (N = 44), n (%N)	Standard Adjuvant TMZ Finished (N = 41), n (%N)	Extended Adjuvant TMZ (N = 42), n (%N)
Median age, years (range)	54 (32–76)	55 (31–80)	55 (20–70)
Gender (M/F ratio)	22 (50%)	20 (48.8%)	24 (57.1%)
ECOG (post-treatment)			
0 1 2	5 (11.4%) 29 (65.9%) 6 (13.6)	7 (17.1%) 23 (56.1%) 7 (17.1%)	7 (16.7%) 28 (66.7%) 7 (16.7%)
Baseline MMSE < 27	4 (3.2%)	6 (4.8%	2 (1.6%)
Corticosteroid use (pre-treatment)	5 (4.0%)	6 (4.8)	3 (2.4%)
Tumor size > 30 mm	16 (26.1%)	17 (24.6%)	22 (31.9%)
Complete resection	14 (31.8%)	14 (34.1%)	19 (45.2%)
RPA class			
III IV V	7 (15.9%) 31 (70.5%) 6 (13.6%)	10 (24.4%) 24 (58.5%) 7 (17.1%)	5 (11.9%) 36 (85.7%) 1 (2.4%)
Standard EBRT protocol	36 (81.8%)	33 (80.5%)	40 (95.2%)
MRI used for EBRT planning	27 (61.4%)	26 (63.4%)	29 (69%)
EBRT finished as prescribed	43 (97.7%)	40 (97.6%)	42 (100%)
Median number of TMZ cycles (range)	3 (1–5)	6 (NA)	12 (7–13)

*List of abbreviations*: TMZ: temozolomide; M/F: male/female; ECOG: Eastern Cooperative Oncology Group; RMMSE: Mini-Mental Status Evaluation; RPA: Recursive Partitioning Analysis; EBRT: external beam radiation therapy; MRI: magnetic resonance imaging.

## Data Availability

The data presented in this study are available on request from the corresponding author. The data are not publicly available due to privacy restrictions.

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
