# Peer review of "Can Extended Chemotherapy Improve Glioblastoma Outcomes? A Retrospective Analysis of Survival in Real-World Patients"

_jpm, 2022, doi:10.3390/jpm12101670_

Round 1

Reviewer 1 Report

I reviewed the manuscript which brought a retrograde cohort studies on GBM. Authors tried to evaluate the effect of extensive chemotherapy by TMZ on survival, performance and mental functions. However, severe concerned raised with this report as following:

- line 35, It is better to say "standard treatment for operable GBM is >>>".because some type of GBM especially those located in the brain stem and thalamus are not resectable. So , in this situation the first line of treatment will be chemo and EBRT.

- what was the criteria  for selected patients who received extensive TMZ?

- The total number of patients in this study should be mention in material section.

- section 3.1 ECOG and MMSE score severely tight to the site of GBM. If the GBM site was in PFC, definitely we expect to face with the cognition deficit. Thus i expect to see analysis based on stratification of patients in different aspect.

Another example ; table 2

Among 79.4% of patients that received TMZ, there are threee form of treatment >6,6, <6. we can not drawn tumor size results as reported.

Thus, report should be based on stratification and correlation analysis between data, such as tumor size and cycle of TMZ or tumor site with mental state.

Author Response

First, we would like to thank the reviewer for his insightful comments and observations. We revised the manuscript accordingly and the changes we made are marked with the track changes option in the revised manuscript uploaded. Below is a point-by-point response to the comments of Reviewer 1.  

  1. Line 35, It is better to say "standard treatment for operable GBM is >>>".because some type of GBM especially those located in the brain stem and thalamus are not resectable. So , in this situation the first line of treatment will be chemo and EBRT.

Thank you for your kind suggestion. We have revised the manuscript accordingly.

  1. What was the criteria  for selected patients who received extensive TMZ?

Since this was a retrospective study, the decision to extend adjuvant TMZ was made by the treating oncologist together with the patient on a case-by-case basis. General criteria used were: good performance status (ECOG PS 0-1), no new neurological symptoms, no need for corticosteroids, good treatment tolerance (hematologic, gastro-intestinal), no clinically significant disease progression.  

Thank you for pointing out this omission. We added the relevant criteria for group definition in the text (section 3.4).

  1. The total number of patients in this study should be mention in material section.

Thank you for your suggestion. We have revised the manuscript accordingly.

  1. section 3.1 ECOG and MMSE score severely tight to the site of GBM. If the GBM site was in PFC, definitely we expect to face with the cognition deficit. Thus i expect to see analysis based on stratification of patients in different aspect. Another example ; table 2 - among 79.4% of patients that received TMZ, there are threee form of treatment >6,6, <6. we can not drawn tumor size results as reported. Thus, report should be based on stratification and correlation analysis between data, such as tumor size and cycle of TMZ or tumor site with mental state.

We thank the reviewer for this excellent suggestion. We had indeed performed a statistical analysis of patients based on tumour location and size, controlling for OS. There was a significant correlation between performance status, cognitive dysfunction and corticosteroid use, but these parameters did not correlate with tumour characteristics or number of chemotherapy cycles. Regarding the relationship between tumour localization and cognitive dysfunction, we only found a trend of inverse correlation (p = 0.075). As such, and due to the small sample size, we did not perform a stratified analysis and we focused on reporting aggregate results. Please find attached an excerpt of the statistical analysis attached.

Reviewer 2 Report

The retrospective observational study is described in the paper. The question of extended TMZ chemotherapy for glioblastoma patients was raised. Although taking into account the retrospective nature of the study, no randomization of the patients can be performed, so no meaningful conclusion can be drawn. The survival data is presented, that increase the value of this study, however absence of MGMT status makes this data more historic.

Author Response

Thank you for your comment. We wholeheartedly agree that the absence of MGMT status is a limitation of our retrospective analysis, which we acknowledged in the Discussion section. However, we decided to report the data as is, since we considered this relatively large series of consecutive GBM patients with data available for long-term survival analysis to be useful for clinical practice.

Additionally, another focus of our analysis was to determine a potential benefit in selected patients for extending TMZ adjuvant therapy beyond 6 cycles. MGMT methylation status can help physicians in making a decision about offering adjuvant TMZ and has been suggested for use as an independent prognostic factor in clinical practice; however, it remains complementary to more well-established prognostic factors (which we did include in our analysis). Last, but not least, this exploratory analysis represents the initial testing background for the design of a prospective randomized clinical trial aiming to detect a potential survival advantage for extending adjuvant TMZ in selected patients.

Round 2

Reviewer 1 Report

Unable to do stratification in this study is one of the major problem which may affect the drawn conclusion. As the authors improved part of the manuscript and added more detail however, they can not recruit more patients, the revised manuscript still is week. But for general interest may be useful.